# Electrical Event Detection and Monitoring Data Storage from Wide Area Measurement System

Vinicius Tertulino Parede [1], Alexandre Rasi Aoki [1], Mateus Duarte Teixeira [1], Thelma S. Piazza Fernandes [1,*], Nathan Elias Maruch Barreto [1], Flavio Lori Grando [2], Vanderlei Aparecido da Silva [2], Fabio Alessandro Guerra [2], Milton Pires Ramos [2], Clayton Hilgemberg da Costa [2], Bruna Machado Mulinari [2], Germano Lambert-Torres [2], Ricardo Rodrigues de Almeida [3], Rafael Rodrigues [3], Victor Frederico Müller Junior [3] and André Katayama dos Santos [3]

[1] Department of Electrical Engineering, Federal University of Parana, Curitiba 81531-990, PR, Brazil
[2] Institute Gnarus, Itajubá 81070-190, MG, Brazil
[3] Copel Generation and Transmission S.A, Curitiba 81200-240, PR, Brazil
* Correspondence: thelma.fernandes@ufpr.br

**Abstract:** Synchronized phasor measurement systems are being widely used around the world and have become essential elements in the evolution of the operation of large electrical power systems (EPS). These systems, called Phasor Measurement Units (PMUs), are capable of recording and communicating dynamic data from the EPSs in a synchronized way by GPS and with a high sampling rate, generate a huge set of data that, among many applications, has the capacity to detect events. In this way, this work presents a data management system architecture applied to a real PMU system located in the state of Paraná, Brazil that detects and storages events using principal component analysis and Pearson correlation. This method can detect and store electrical events that occurred during the operation of the national interconnected system of Brazil with good results.

**Keywords:** principal component analysis; wide-area monitoring; phasor measurement units; electrical events detection

## 1. Introduction

Synchronized phasor measurement systems are essential elements in the evolution of the operation of large electrical power systems (EPS). These systems are based on Phasor Measurement Units (PMU), which can record and communicate data on the dynamics of the EPSs in a synchronized way by GPS and with a high sampling rate, composed of up to 60 samples per second.

The importance of an integrated PMU system is that it permits to observe the behavior of the all the EPS, monitor large areas, support the stability supervision of transmission systems, observe the dynamic behavior of the electrical network, monitor the conditions in real time and realize offline studies.

Over the past decade in the United States, there has been a continental grid with more than 2000 PMUs helping to improve the reliability of the North American electric power grid [1]. Likewise, in China, there is already a wide network with more than 2400 PMUs covering all 500 kV substations in the country and several important power plants and substations of 220 kV [2].

In continental Europe, there is a phasor measurement system covering almost the entire territorial area of the different countries. Applications with data exchanged between different operators of transmission systems allow them to manage and operate the entire system safely and efficiently [3].

In Brazil, the implementation of a synchronized phasor measurement systems has also been growing, following the coordination and standards established by the ONS—National Electric System Operator. In this regard, the state of Paraná, through COPEL Generation

and Transmission S.A., was a pioneer in the implementation of the largest system in the country, whose data are used in this work.

The Wide Area Measurement System (WAMS) of Copel is composed by 63 phasor measurement units installed around substations of 230 kV and 525 kV, Phasor Data Concentrators (PDC) and a system that enables communication between them.

The 63 PMUs of the Copel WAMS permit the visualization of data in real time and an effective way to perform analysis and operation of its own electrical network and of the Brazil's interconnected system in which Copel's system is inserted. Then, the influence of events that occur along anywhere of Brazil's interconnected system can be monitored, by applications for real-time and offline studies [4], making better the supervision of the electrical system of Brazil.

However, as the amount of data generated by PMUs is very large, the concept of data analytics has become fundamental to extract useful information from PMUs and thus generate, for example, historical data of anomalies, which can help companies performing post-operation analysis, EPS planning, managing indicators and forecasting trends [5].

Thus, in this article, data analytics will be used to treat the voltage, current, angles and frequency data generated at each cycle by the PMUs installed in the electrical system of the state of Paraná, in Brazil, to provide useful data that can be used by the company for signaling and post-operation management.

The literature presents some data analytics applications for anomaly detection:

- Ref. [6] performs an analysis on the dimensionality of the PMU data for both normal and abnormal conditions, using an algorithm based on the changes detected within the subspace created by the dimensionality reduction.
- Ref. [7] describes a method based on Principal Component Analysis (PCA) capable to locates electrical power system faults exposed to different types of disturbances by combining the input data of phasor synchronous meters.
- Ref. [8] presents methods to detect events and storage reduction data using Principal Component Analysis (PCA) method with a second-order differential method. The proposed method for data reduction is based on an event-driven and self-adjusting sliding window.
- Ref. [9] uses Random Matrix Theory (RMT) as data processing tools to estimate the state of large power systems. The developed algorithm performs a high-dimensional analysis and compares it with the RMT predictions for anomaly detections in the electrical system.
- Ref. [10] proposes a dedicated method based on rules for events detection, such as monitoring normal operating limits.
- Ref. [11] obtains data from PMUs in a reduced form using the local outlier factor algorithm to detect and locate events.
- Ref. [12] detects and locates single-phase-to-ground faults by correlating the values of electrical quantities and the status of the power system.
- Ref. [13] presents a PMU anomaly detection that classifies events, outliers, and the lack of measurements. This system is based on stacking machine learning techniques to obtain a higher level of accuracy and increased performance with high-dimensional data. After capturing data from PMUs, the isolation forest technique is applied, which provides scores that classify the data as normal or anomalous (which are the events). These scores feed two other K-Means and LoOP techniques, whose results are multiplied vectorially, and which result in probabilities that are applied to Pearson's correlation with other PMUs to verify whether an event is occurring.
- Ref. [14] presents a convolutional neural network (CNN)-based model to detect frequency disturbance events.
- Ref. [15] exploits the statistic properties of the PMU dataset and generate a hypothesis testing framework to detect power system events using sample covariance of the PMU data collected during the system operations.

- Ref. [16] uses neural network-based event detection and classification algorithms that requires thousands of confirmed events as training labels.
- Ref. [17] develops a bidirectional anomaly generative adversarial network (GAN) algorithm to detect power system events with the introduction of conditional entropy constraint in the objective function of GAN and graph signal processing-based PMU sorting technique.

Some considerations can be made about the literature studied: works based on signal processing techniques (wavelet transforms and Fourier transforms, for example) do not fully exploit the spatial correlations between data from different PMUs [17]; the works that exploit the statistic properties of the PMU dataset to detect events, have complexity of correlation matrix calculations [17]; works that uses data mining techniques to detect power system incurs high computation cost for real-time applications [17] and the ones that use neural network-based event detection require thousands of confirmed events as training labels [17].

However, the technique proposed in this work, the PCA (already well known in others power system applications), does not expend high computation cost and can be easily implemented with good results, as will be described.

Through the synthesis of the selected technical-scientific articles, the following conclusions can be drawn:

- the techniques usually used for event detection are: principal component analysis, state extraction method, non-nested generalized examples; random matrix, isolation forest, K-means, LoOP, among others
- most works consider application of a centralized approach to control
- few works consider real-time application aspects of real systems
- most of the data analytics techniques are being used in the analysis of electricity distribution problems.

Therefore, the present article contains as contributions the use of a distributed approach for control, application in real time for large electrical systems (transmission) of a Brazilian utility. For that, the PCA technique [8] is used, together with the Pearson Correlation whose results were compared with the results of [13], to indicate potential power grid events.

So, the data processing proposed carried out on the intelligent platform aims to detect in real time from an amount of PMUs measurements, anomalous situations of the electrical network and the measurement system itself. These detections include:

(i)   status detection: PMUs have an algorithm that generates information about the device's status, which may indicate data error, PMU error, modified data, loss of satellite communication, among others
(ii)  violations of operational limits
(iii) finally, the application of PCA technique and Pearson correlation to monitor measurement values and detect violations of operational thresholds

All these sets automatically trigger data storage, enabling the maintenance of a history of electrical network occurrences.

## 2. Description of PMU System of State of Paraná (Brazil)

Copel Generation and Transmission (Copel GeT) has PMUs installed in all its substations, power plants and transmission lines. Most of the devices are in the state of Paraná and some outside, such as in São Paulo (bordering the state of Paraná), Santa Catarina (bordering the state of Paraná), Mato Grosso (located more than 2000 km from Paraná) and Rio Grande do Norte (located more than 3000 km from Paraná). Altogether, the system comprises an increasing number of 63 devices, each with dozens of measurement channels.

Copel GeT has PMUs so electrically distant, as Mato Grosso and Rio Grande do Norte, because it has power plants installed at those states.

Figure 1 shows the main screen of the software used by Copel GeT [18] which enables the viewing of its PMUs located in Copel's concession area (green diamonds) and in neighboring states.

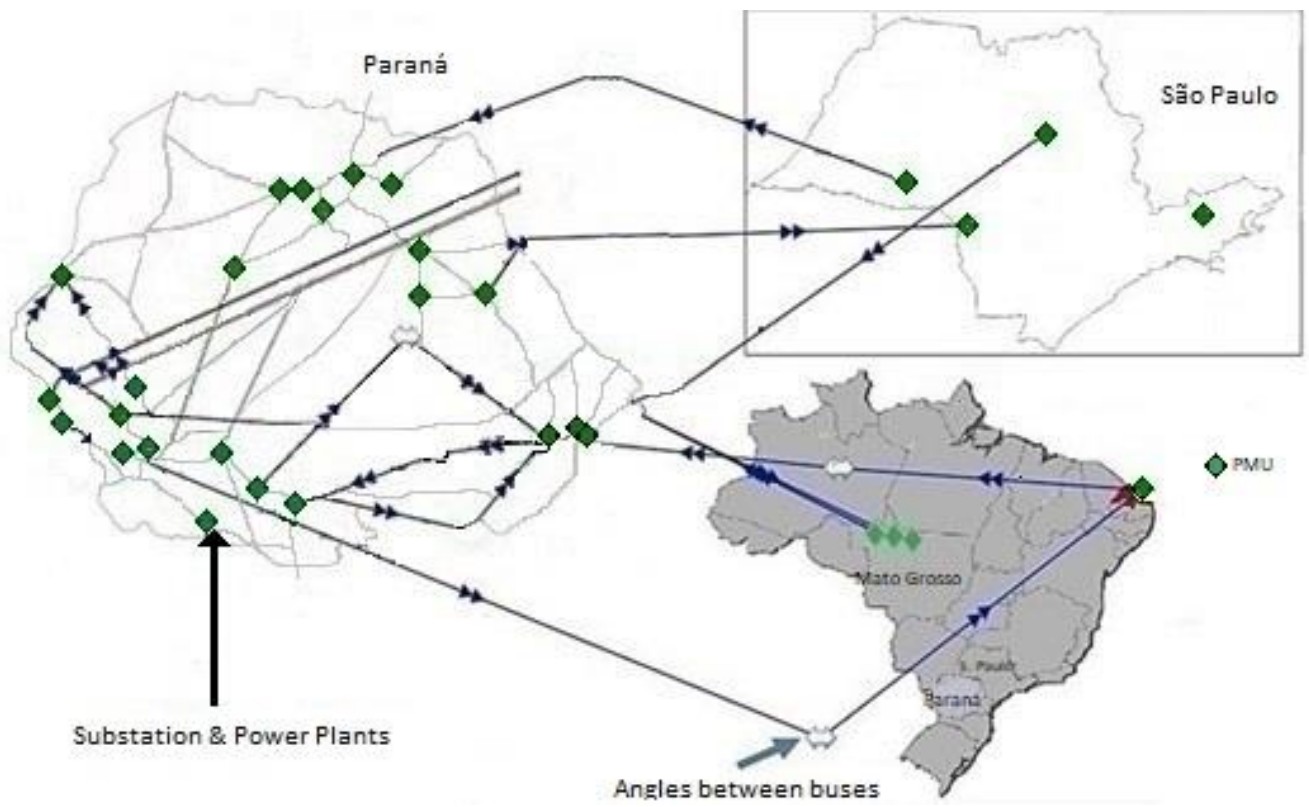

**Figure 1.** Localization of PMUs (green diamonds)—Copel GeT (adapted from [18]).

In this way, Copel WAMS produce thousands of pieces of information which are updated dozens of times per second. Despite the benefits, the high volume of data produced makes manual analysis unfeasible, requiring sophisticated systems to manipulate the amount of information.

To analyze these questions, a research project was developed with the ANEEL Research and Development Program, Copel Generation and Transmission, Federal University of Paraná (UFPR) and the GNARUS Institute that proposed a tool capable of handling, in real time, the large volume of data coming from Copel GeT PMU network.

The development of a computational platform to process data from PMUs has special characteristics, whose challenges are large volume of information, diversity (types) of information and high refresh rate.

The purpose of the platform proposed is to deliver useful information to the operating sector, which means that the system must have the following characteristics: fast response (to support real-time operation) and data storage of only electrical events.

The measurements captured from all the PMUs are transmitted through frames following the IEEE C37.118 protocol [19,20]. Frames are vectors which bring information about the communication and carry the data. PMUs send up to 60 frames per second and they are uninterrupted. Therefore, featuring a data stream updated once every 16.6 milliseconds.

For that, a computational platform was proposed with a set of fundamental functionalities, such as: communication with PDC, data processing, storage (database), real-time visualization, report generation and, finally, support and management of all modules.

Communication between computers can use both TCP (Transmission Control Protocol) and UDP (User Datagram Protocol). In the first case, there is the control of data packets during a transmission, offering retransmission in case of loss or error in the packets. UDP,

on the other hand, is a simpler protocol and does not allow re-transmission. On the other hand, it is faster and with greater bandwidth, therefore, it allows transport of greater volume of data. The messages are defined by the IEEE C37.118 protocol [19,20], performed by a typical communication architecture.

The technology used for data storage is National Instruments (NI)'s native solution for streaming data. NI introduced the Technical Data Management System (TDMS) file format in LabVIEW 8.2 to overcome some shortcomings of other data storage options commonly used in test and measurement applications.

The data processing analyzes the number of measurements of the electrical network. This step involves a set of detectors developed to identify and report situations in the electrical network and in the measurement system itself.

One of these detectors will be described in the next section.

Figure 1 shows, the example of the main screen of the PDC software used in Copel GeT.

## 3. Data Anomaly Detection Using Principal Component Analysis

Copel Generation and Transmission has PMUs installed in all its substations, power plants and transmission lines. Most of the devices are in the state of Paraná and some outside, such as in São Paulo, Santa Catarina, Mato Grosso and Rio Grande do Norte estates. Altogether, the system comprises an increasing number of 63 devices, each with dozens of measurement channels. This section aims to describe assumptions and procedures adopted for the detection of events (or anomalies) monitored by PMUs, which use simple monitoring rules together with the principal component analysis technique and Pearson's correlation.

PCA is one of the most widely used and useful tools in the field of exploratory data analysis because it offers an overview of the subject in question, showing the relationship that exists between objects, as well as between objects and variables [21]. It allows the data to maintain its original structure, making only an orthogonal rotation of the variables, which helps to simplify the visualization of all the information already contained in the data [21].

Pearson's linear correlation coefficient is a commonly known method for measuring the correlation between several variables. This correlation coefficient, R, measures the statistical relationship between continuous variables [22]. Thus, (i) if R < 0 it means that the variables are inversely related; (ii) if R = −1 means a perfect negative correlation between the variables, that is, if one increases, the other always decreases; (iii) if R > 0: it means that the variables are directly correlated, (iv) if R = 1 it means a perfect positive correlation between the two variables; and (v) if R = 0, it means that it is not possible to determine any sense of covariation.

So, as close as the values of R are to the extremes, the stronger is the correlation. As further the values are to the middle, or the closer to zero, the weaker it becomes (no correlation).

Thus, in possession of these two powerful tools, the proposed method for event detection in ESP is described step by step, and are repeated (from step 1 to step 8) for each tumbling window of 1800 samples (for example) and for each datapoint of the PMU being considered.

The data from the PMUs are basically composed of:

- Timestamps: date (day, month, year) and time (hour, minute, second and thousandth of a second).
- Meter status: data error, GPS signal, ordering, triggers, configuration change, modified data, timing quality, synchronism.
- Electrical system data: frequency, rate of change of frequency, voltage phasors (magnitude and angle), current phasors (magnitude and angle), analogue and digital channels.

This information is present in all PMUs and is transmitted through frames following the IEEE C37.118 protocol. The frames are vectors which bring information about the communication and also carry the data mentioned above. PMUs send up to 60 frames per second and non-stop. Therefore, featuring a stream of data updated once every 16.6 milliseconds.

So, for each tumbling window of 1800 samples, for example, captured by the client-server communication architecture foreseen in the IEEE C37.118.2 protocol for each PMU, the follow steps are described:

Step 1: treatment of the measured data for each of the quantities made provide by the PMU. It consists of removing columns referring to date and time and applying IEEE C37.118 protocol, using bits 14 and 15, to detect if the measured data is valid or not.

These detections identify numerical problems such as absence of values (Not a Number—NaN), null values, poor data quality, which are reporting by rates less than 60 frames per second. This functionality also generates a report reporting problems identified in each measurement channel.

When invalid data is detected, it is replaced by the previous sample, to ensure synchronization in the timestamp of all PMUs.

The Figure 2 illustrates the capture of the *m* = six measurements (frequency, rate of change of frequency, magnitude, and angle of voltage and current of phase A) during 100 samples of a PMU.

| | Frequency | df/dt | VA1 V_CVN_230kV Magnitude | VA1 V_CVN_230kV Angle | IA1 I_CVN_230kV Magnitude | IA1 I_CVN_230kV Angle |
|---|---|---|---|---|---|---|
| 0 | 59.988293 | 0.002908 | 236708.53125 | −24.606995 | 49.700466 | 150.578262 |
| 1 | 59.988338 | 0.002894 | 236708.34375 | −24.674858 | 49.699326 | 150.509781 |
| 2 | 59.98838 | 0.002879 | 236708.15625 | −24.74272 | 49.698185 | 150.441299 |
| 3 | 59.988422 | 0.002864 | 236707.9375 | −24.810581 | 49.697041 | 150.372833 |
| 4 | 59.988468 | 0.002849 | 236707.75 | −24.878443 | 49.6959 | 150.304352 |
| .. | ... | ... | ... | ... | ... | ... |
| 95 | 59.992622 | 0.002266 | 236729.125 | −30.149261 | 49.734428 | 145.197311 |
| 96 | 59.992668 | 0.002265 | 236729.640625 | −30.200676 | 49.735878 | 145.149261 |
| 97 | 59.992718 | 0.002264 | 236730.15625 | −30.252089 | 49.737324 | 145.101242 |
| 98 | 59.992764 | 0.002263 | 236730.703125 | −30.303503 | 49.738773 | 145.053207 |
| 99 | 59.992813 | 0.002263 | 236731.203125 | −30.354918 | 49.740219 | 145.005157 |

**Figure 2.** Data of frequency, rate of change of frequency (df/dt), magnitude and angle of voltage and current of phase A).

Step 2: identification of measurements outside the operational limits normalized by the operator system (ONS). In this way, when frequency quantities or voltage magnitudes are outside their normative limits, the recording of the event begins regardless of the application of the PCA technique.

The data are stored for frequency signals with values above 60.5 Hz and below 59.5 Hz and for voltage magnitude values outside the limits of 0.95 pu and 1.05 pu.

Step 3: selection of the normal operating range. For the tumbling window being analyzed (1800 samples) and each monitored quantity, mean and standard deviation are calculated for this range.

The mean and standard deviation values are calculated for each operating period (for example, 1800 data or 30 s). This strategy is adopted to learn what is a normal or not operation comportment. If larger interval of data that could contain an event is considered, these mean and standard deviation would not represent a reference to a normal operating pattern, because the deviations caused by the event would be embedded inside the measurement window.

Step 4: standardization of the data contained (for the tumbling window and for each measurement of the PMU being analyzed, represented by X) inside the pre-selected range from Step 3, according to the previously calculated average and standard deviation.

$$X = \frac{average}{standard\ deviation} \tag{1}$$

Figure 3 illustrates the normalization of the six measurements (frequency, rate of change of frequency, magnitude, and angle of voltage and current of phase A) during 100 instances of time of a PMU from Figure 2.

| | Frequency | df/dt | VA1 V_CVN_230kV Magnitude | VA1 V_CVN_230kV Angle | IA1 I_CVN_230kV Magnitude | IA1 I_CVN_230kV Angle |
|---|---|---|---|---|---|---|
| 0 | 0.999805 | 0.002908 | 0.450873 | −0.429473 | 0.45194 | 2.628086 |
| 1 | 0.999806 | 0.002894 | 0.450873 | −0.430658 | 0.451929 | 2.626891 |
| 2 | 0.999806 | 0.002879 | 0.450873 | −0.431842 | 0.451919 | 2.625696 |
| 3 | 0.999807 | 0.002864 | 0.450872 | −0.433026 | 0.451908 | 2.624501 |
| 4 | 0.999808 | 0.002849 | 0.450872 | −0.434211 | 0.451898 | 2.623306 |
| .. | ... | ... | ... | ... | ... | ... |
| 95 | 0.999877 | 0.002266 | 0.450913 | −0.526204 | 0.452248 | 2.534171 |
| 96 | 0.999878 | 0.002265 | 0.450914 | −0.527101 | 0.452262 | 2.533333 |
| 97 | 0.999879 | 0.002264 | 0.450915 | −0.527999 | 0.452275 | 2.532494 |
| 98 | 0.999879 | 0.002263 | 0.450916 | −0.528896 | 0.452288 | 2.531656 |
| 99 | 0.99988 | 0.002263 | 0.450917 | −0.529793 | 0.452301 | 2.530817 |

**Figure 3.** Data of frequency, rate of change of frequency (df/dt), magnitude and angle of voltage and current of phase A, normalized.

Step 5: calculation of PCA [21], based on the standardized values (X) of all measured quantities (from Step 4), for each PMU of the tumbling window considered.

Regarding the calculation of the PCA, firstly it is calculated for each instance (for example, 100 instances of Figure 2) of the covariance matrix C (with dimension $m \times m$, or $6 \times 6$, as Figure 2). After having obtained the covariance matrix, the eigenvalues $(\lambda_1, \lambda_2, \ldots, \lambda_m)$ and the eigenvectors $(a_1, a_2, \ldots, a_m)$ must be identified.

Having identified the eigenvalues and eigenvectors, the calculation of the principal components itself is carried out. It is made by a linear combination of the $m$ measurements being weigh by the eigenvectors, providing the principal component values, $(P_1, P_2, \ldots, P_m)$, for each of the instances of the data set. Thus, a list of $m$ principal components, for each instance is obtained.

Finally, to reduce the number of PCAs, this work uses only one PCA, that includes the one with the greater eigenvalues.

Step 6: for the PCA calculated (that includes information about all the measurements of the PMU for each instance), the mean and standard deviation are again obtained, and the Z-score is calculated (Equation (1)). This is performed so the larger scale quantities do not dominate the process over the others; in this way they can be observed within the same scale range. Thus, a standardized vector is assembled, whose columns are the PCA for each instance.

Step 7: after obtaining the z-score values, all peaks of the PCA quantity that are above pre-specified values are detected as events. The limits used for the Z-core values are $\pm 3$ deviation above the mean of the PCA. That is, values outside the limits of $\pm 3$ deviation are detected as events and they are considered as unitary values, while the values inside the limits are considered as null values.

Steps 1 to 7 are repeated for each one of the PMUs, then Step 8 correlates the detection of all the PMUS obtained.

Step 8: application of Pearson's correlation, considering all quantities treated from Steps 1 to Step 7. That is, the quantities detected as events individually to each PMU are now correlated between all of them to definitively conclude about the event occurrence.

In this article, the values suggested in [21] are used to classify the correlations between the quantities monitored by the PMUs, after applying the PCA of each detected value inside the interval the tumbling window. The classification rule is [22]: if R = |0.30| there is a weak correlation; if R = |0.50| there is a moderate correlation; and if R = |0.70| there is a strong correlation.

Step 9: for all the interval monitored, all the measurements contained inside the interval composed by the union of events detected by the PMUs (according to Step 8) are stored.

The flowchart in Figure 4 illustrates the sequence from Steps 1 to 9 that were described.

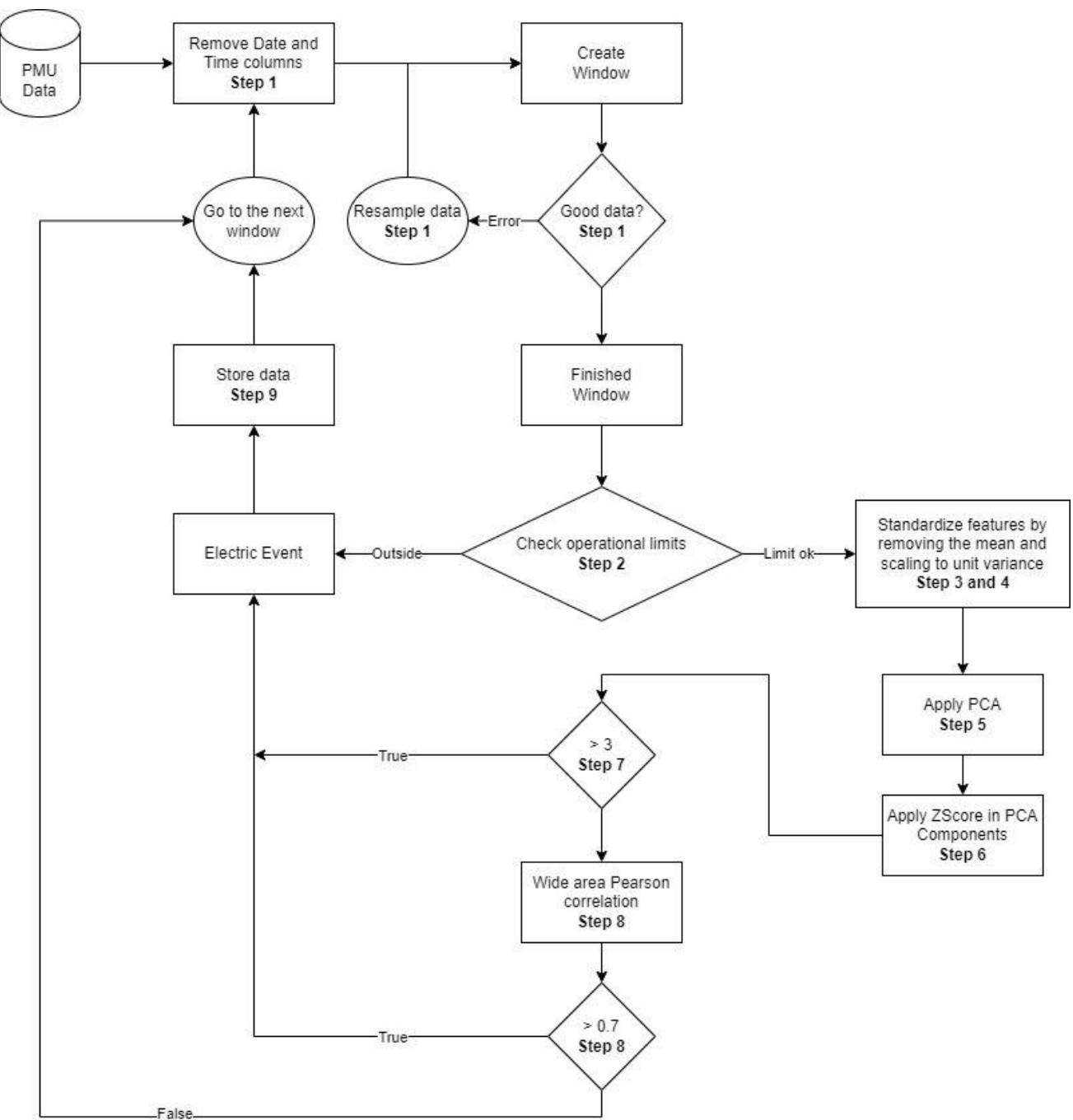

**Figure 4.** Method to detect electrical events.

The standardization is realized by Steps 1 to 4; the PCA is applied by the Step 5; the Z-score application by Steps 6; the application of the limit of 3 is realized by Step 7 and the Pearson correlation by Step 8.

After the detection points, they are stored by Step 9.

## 4. Results

The method described in the previous section was tested using data from two electrical events:

Event 1: the event took place on 28 May 2021, caused by automatic disconnection of pole 1 of the Direct Current Link (DCL). This event had a great impact on the Brazilian network; more details will be described.

Event 2: the event took place on 23 October 2021, caused by a strong storm in the western region of Paraná, which caused the shutdown of the 750 kV Transmission Lines that connects the Itaipu Binacional.

### 4.1. Study of Event 1

On 28 May 2021, at 11:06 a.m., pole 1 of DLC 800 kV Xingu/Estreito with 1996 MW was automatically switched off, with this power being taken over by the remaining poles, without major consequences for the NIS (National Interconnected System).

At 11:26 a.m., pole 2 of DLC 800 kV Xingu/Estreito and seven generating units at UHE Belo Monte (Norte Energia S.A.), which were currently generating an amount of approximately 4050 MW, were automatically shut down.

As a result, there was a reduction in the frequency of the NIS, causing the operation of first stage of the ERAC (Regional Load Relief Scheme), which interrupted about 3400 MW of loads in the NIS.

At 11:31 a.m., the NIS released the recomposition of the interrupted loads, which started at the same time and ended at 11:45 a.m.

This subsection will present the results obtained from the proposed method to detect anomalys. It is applied using the measurements of frequency of PMUs located in Apucarana, Cascavel and Bateias substations (Figures 5–7, respectively) in the state of Paraná over a duration of 1 h, starting at 11:00 p.m. These data were captured during Event 1, which described Apucarana, Cascavel and Bateias substations in the North, West and East of state do Paraná, respectively [1].

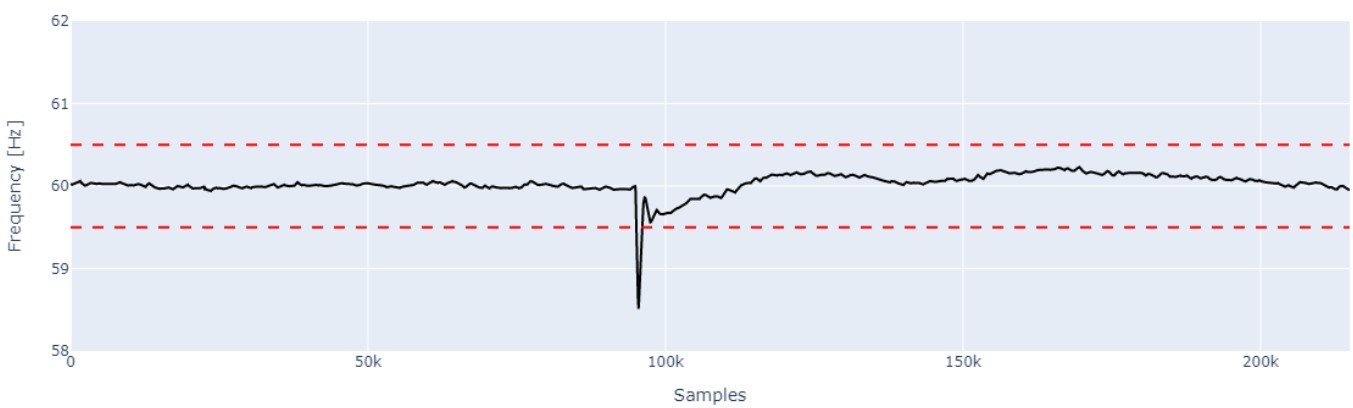

**Figure 5.** Event 1: frequency of PMU Apucarana (in the north of Parana State).

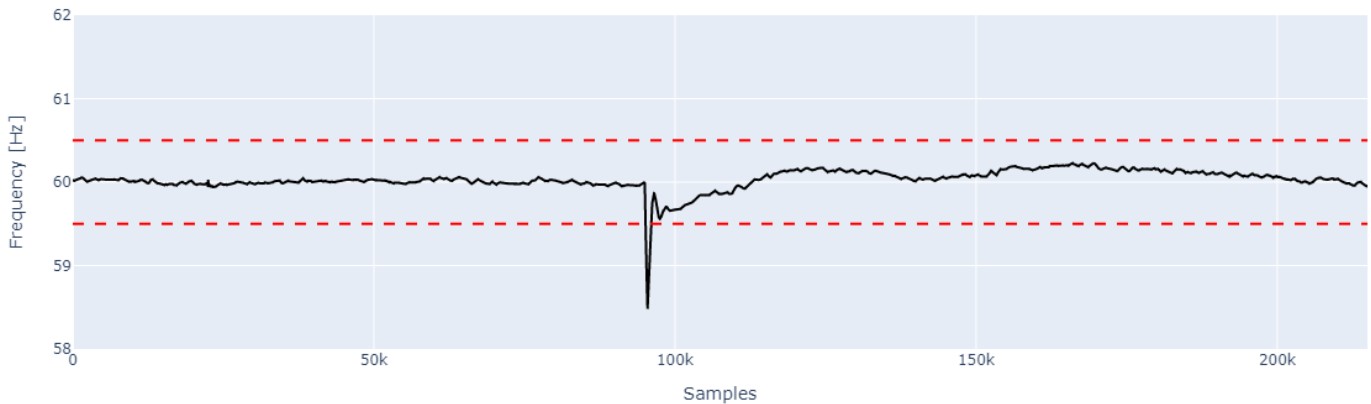

**Figure 6.** Event 1: frequency of PMU Cascavel Oeste (in the west of Parana State).

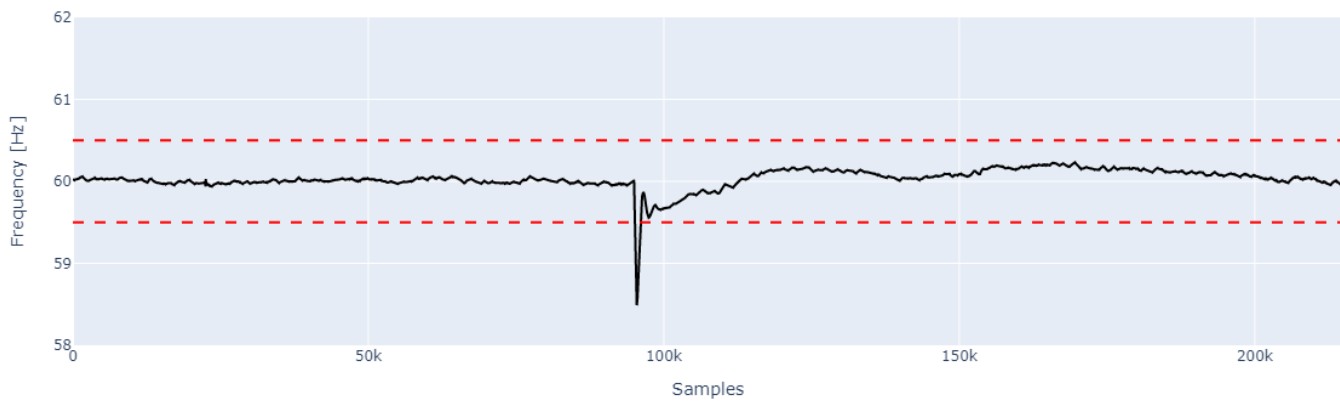

**Figure 7.** Event 1: frequency of PMU Bateias (in the east of Parana State).

Steps 1 to 7 are applied to the measurements of these three PMUs, that is: remotion of measurements not valid (Step 1); identification of measurements outside the operational limits normalized by the operator system (ONS) (Step 2); calculation of mean and standard deviation for each tumbling window (900 samples) (Step 3); standardization of data contained inside the tumbling window (Step 4); calculation of PCA (Step 5); calculation of mean, standard deviation and Z-score (Step 6) and applications of the limits of $\pm 3$ deviation to the Z-core (Step 7). That is, values outside the limits of $\pm 3$ deviation are detected as events and they are considered as unitary values, while the values inside the limits are considered as null values.

After the application of Steps 1 to 7, Figures 8–10 present the signal of frequency, PCA standardized and the points of detection (unit bars) of PMUs located in Apucarana, Cascavel and Bateias substations, respectively.

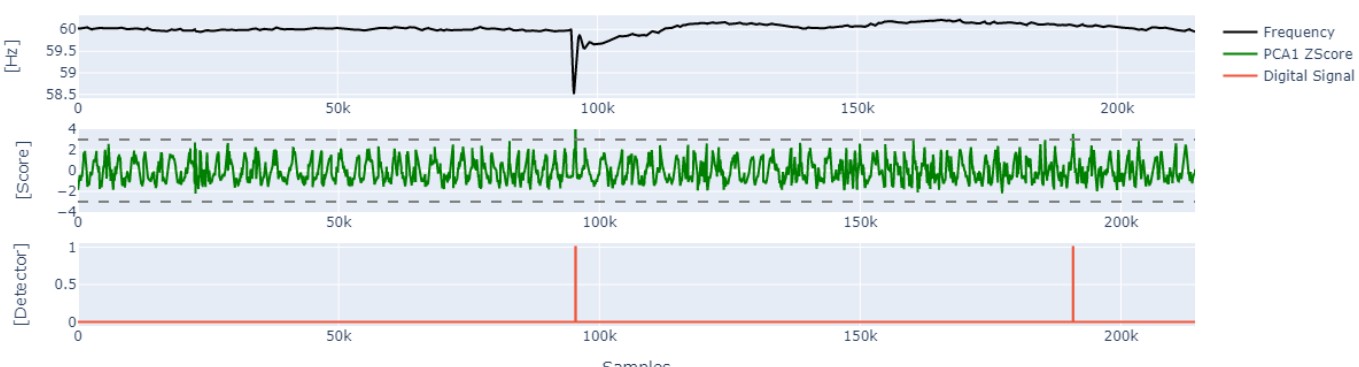

**Figure 8.** Event 1: detector event of PMU Apucarana (located at Parana State).

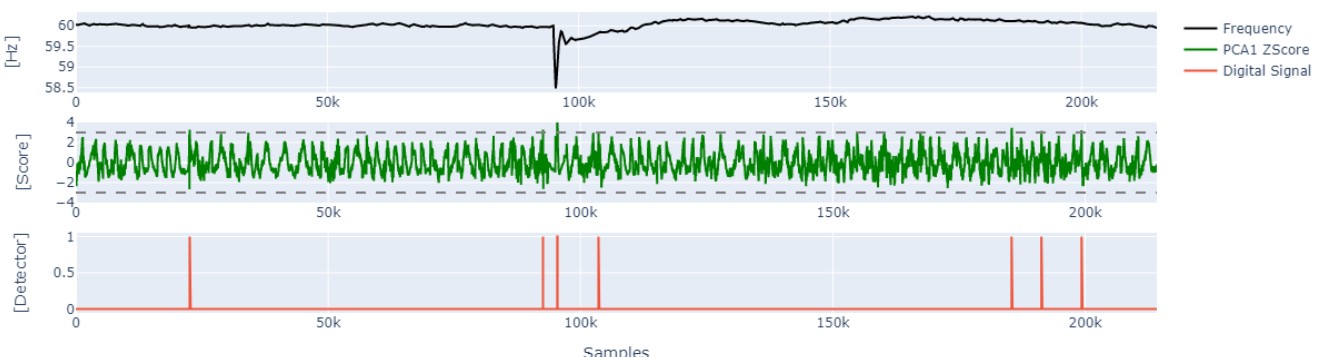

**Figure 9.** Event 1: detector event of PMU Cascavel (located at Parana State).

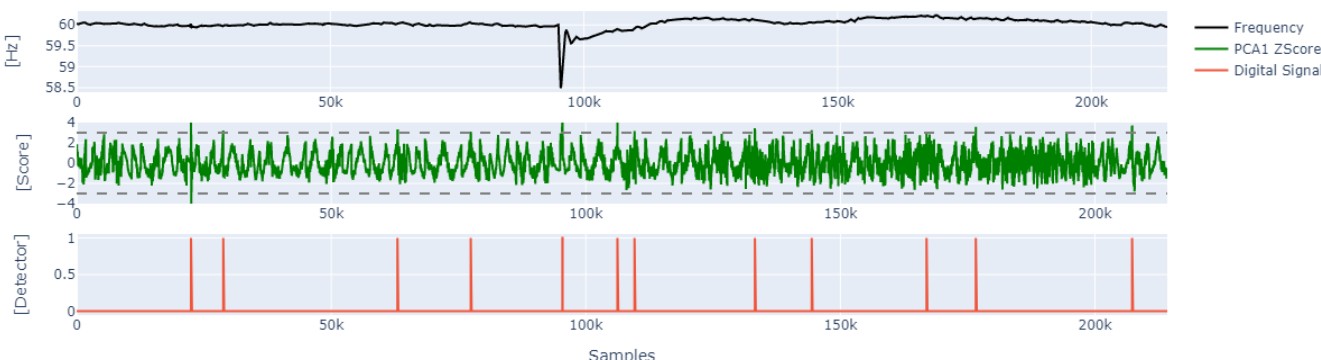

**Figure 10.** Event 1: detector event of PMU Bateias (located at Parana State).

The PCAs standardized are calculation including measures of frequency, rate of change of frequency, magnitude and angle of voltage and current of each PMU.

From these Figures, the PMU Apucarana has two points of detection, Cascavel has seven point of detections and Bateias has 12 points of detection.

The number of points detected depends on the geographic location of the PMUs. For example, Bateias Substation is strategically located in the Paraná network because it is a point of connection with other regions of the state. This fact justifies why its greater reaction to events.

After the application of Step 8, Figure 11 shows the points detections of each PMUs after the application of Pearson correlation (R). Remembering that the classification rule is: if R = | 0.70 | there is a strong correlation, so there is an event.

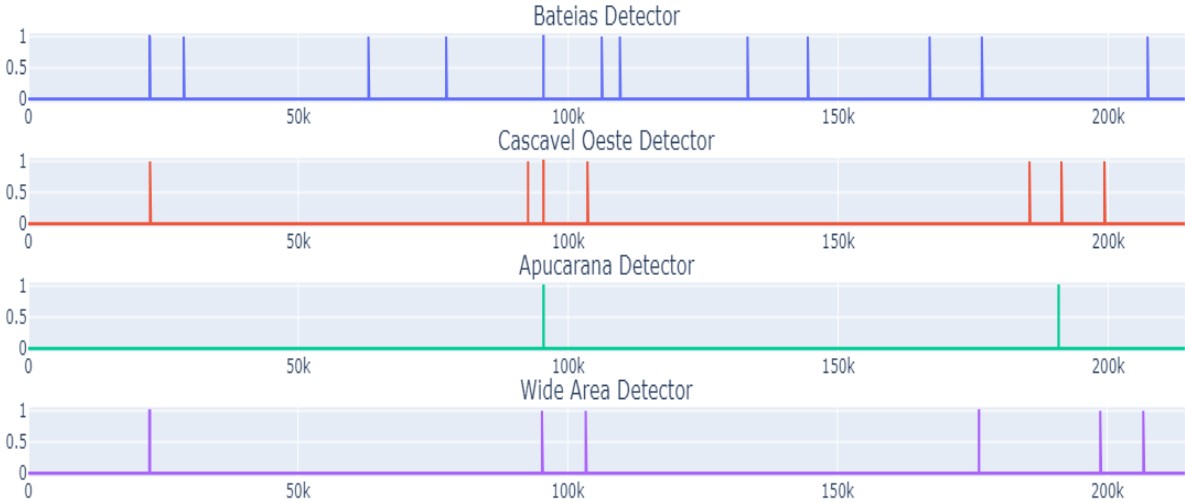

**Figure 11.** Event 1: PMUS detectors and final detector.

At Step 9, it is realized the union of all the events detected by the PMUs (according to Step 8) and they are stored.

The Figure 12 shows the final detection (Step 9) together with the signal of frequency to facilitate the comparations and analysis.

According to Figure 12, the first detection point corresponds to the switched off DLC that occurred at 11:06 a.m., the second one, corresponds to the shutdown generation of 4050 MW at 11:26 a.m., and at 11:31 p.m. begin the recompositing of the system (third point detection. However, an event was detected before, at 11:28 p.m.

So, the technique detects successfully the first two events and detects the third one precociously. The last three points detected are false detections. If only Step 2 was implemented (identification of measurements outside the operational limits), only the event at 11:26 p.m. will be detected and a smaller window of samples could be stored.

The method proposed in [15], which uses the isolation forest, K-means and LoOP techniques was implemented, and it was also able to detect the event illustrated in this work.

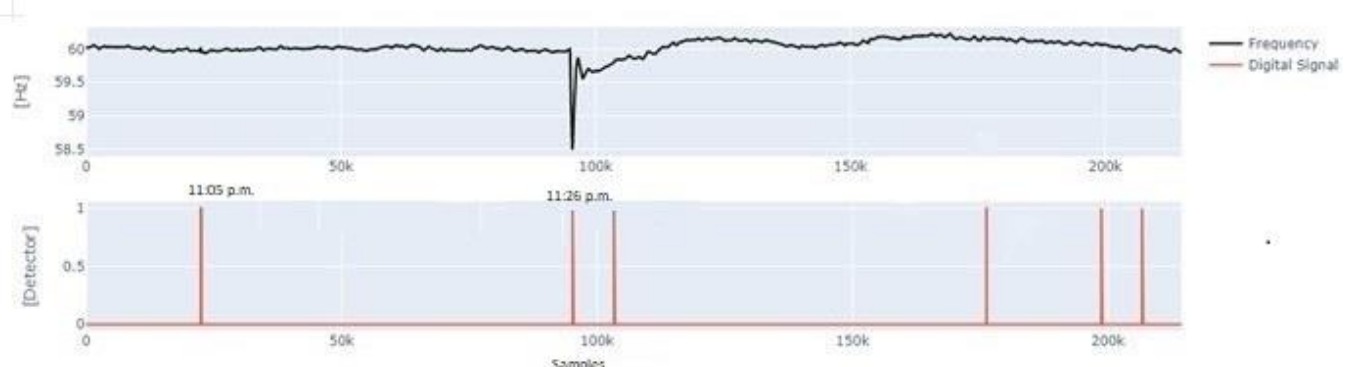

**Figure 12.** Event 1: final detector.

### 4.2. *Study of Event 2*

The event took place on 23 October 2021 at 1:42 p.m. on the power grid of the state of Paraná, Brazil.

This event occurred due to a strong storm in the western region of Paraná, which caused the shutdown of the 750 kV Transmission Lines that connect the Itaipu Binacional plant (with 20 generators) to the rest of the system, and temporary shutdown of five of the ten generating units power plant in the 60 Hz sector and 1635 MW cut-off in units 10, 11 and 17 due to activation of the 765 kV trunk emergency control scheme.

Then, at 1:59 p.m., the 525 kV Cascavel Oeste/Foz do Iguaçu 60 Hz Transmission Line was automatically shut off, leading to the cut of 1200 MW in units 18 and 19. At 2:28 p.m., another automatic shutdown of the 765 kV Line Foz Line occurred. Iguaçu/Ivaiporã cut 900 MW from generating units 14 and 15.

This subsection will present the results obtained from the proposed method to detect the anomaly. It is applied using the measurements of PMUs located:

- in the substations of Copel GeT: Apucarana (in the north of Parana State), Cascavel (in the west of Parana State), Bateias (in the east of Parana State) and Maringa (in the north of Parana State are substations of Copel GeT [1]
- in the Power Station of Copel GeT: José Richa, Ney Braga and Salto Santiago (Paraná) [1]
- in the substations of São Paulo: Araraquara (in the north of São Paulo State), and Itatiba (in the east of São Paulo) [1]

Figures 13–21 show the signal of frequency of the PMUs described, the PCA values and the points detections of each PMU cited, along more than 2 h, starting at 1:20 p.m.

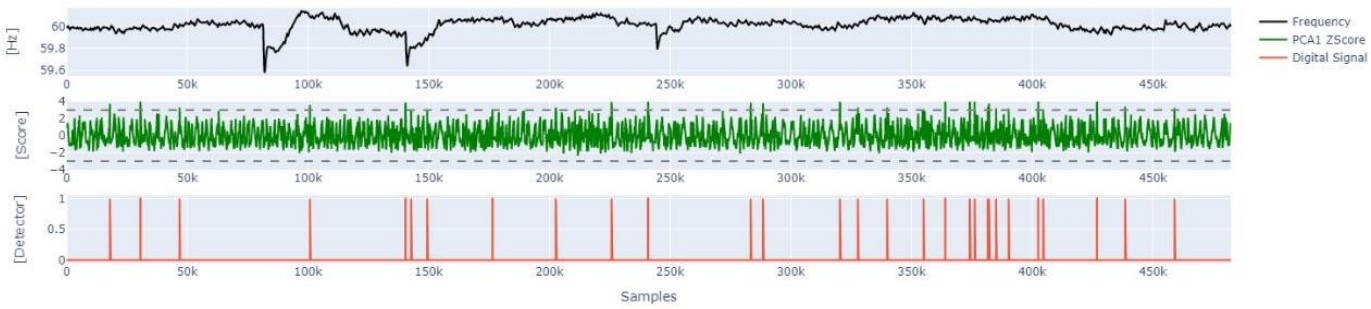

**Figure 13.** Event 2: detector event of PMU Apucarana (located at north of Parana State).

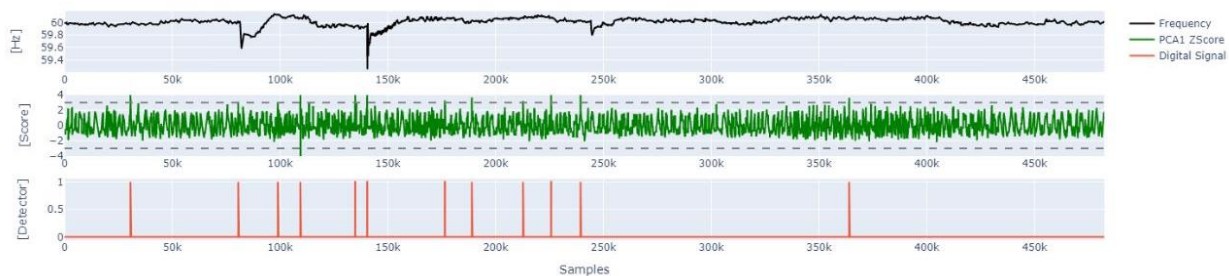

**Figure 14.** Event 2: detector event of PMU Cascavel (located at Parana State).

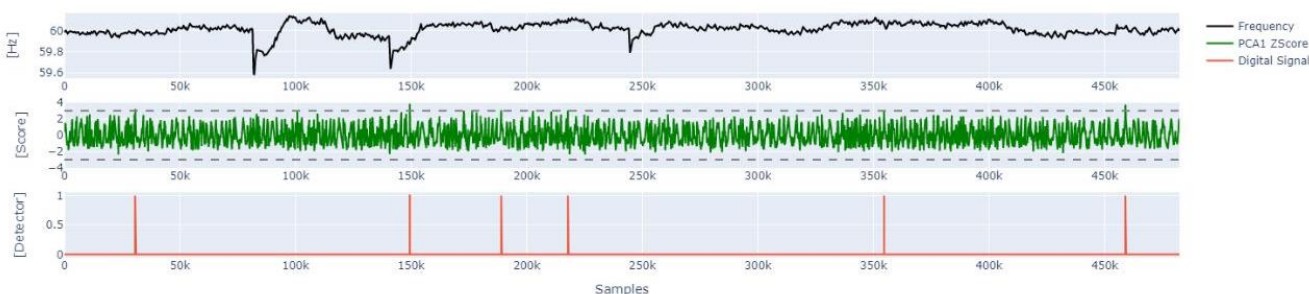

**Figure 15.** Event 2: detector event of PMU Bateias (located at Parana State).

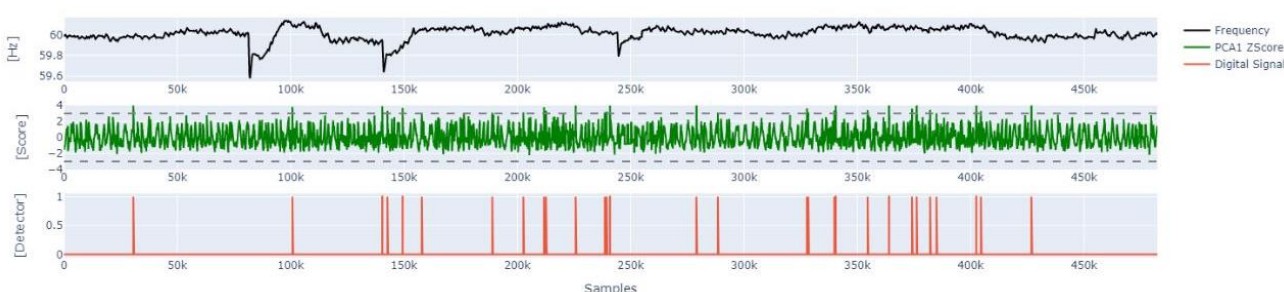

**Figure 16.** Event 2: detector event of PMU Maringa (located at Parana State).

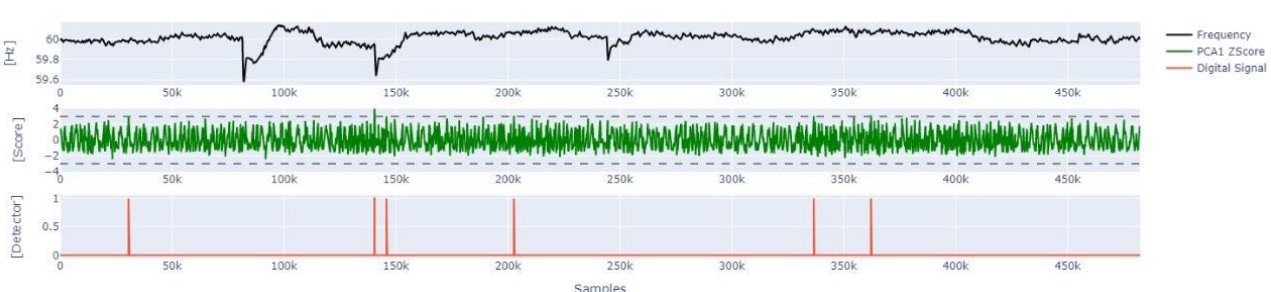

**Figure 17.** Event 2: detector event of PMU Itatiba (located at São Paulo State).

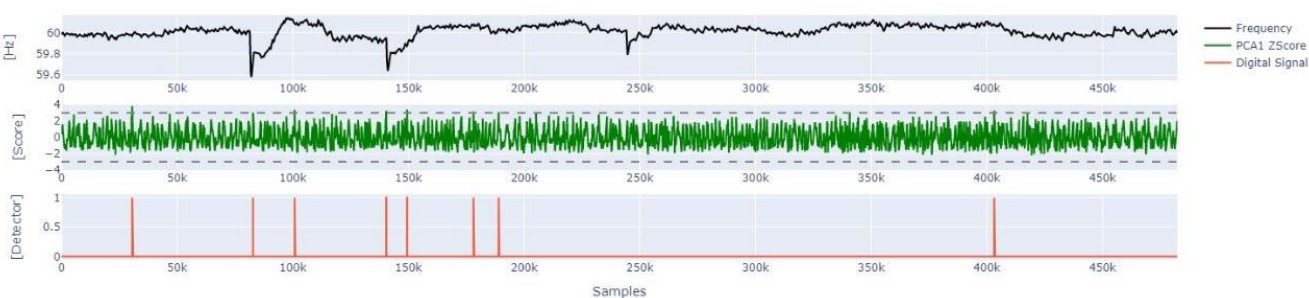

**Figure 18.** Event 2: detector event of PMU Araraquara (located at São Paulo State).

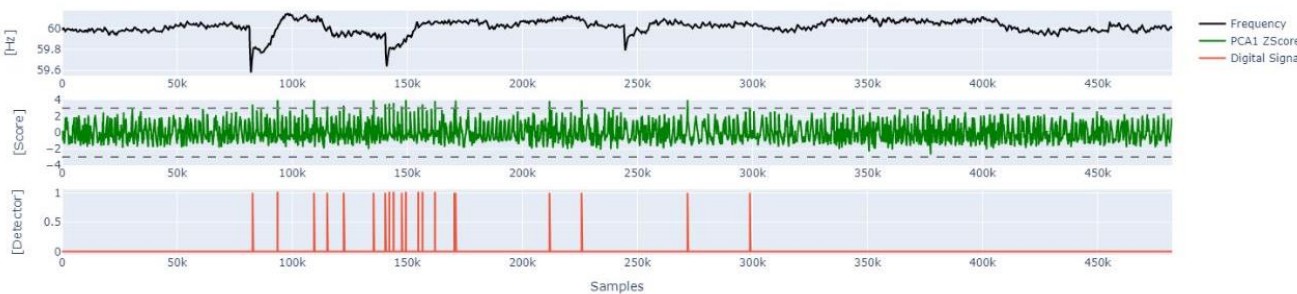

**Figure 19.** Event 2: detector event of PMU hydroelectric power station José Richa.

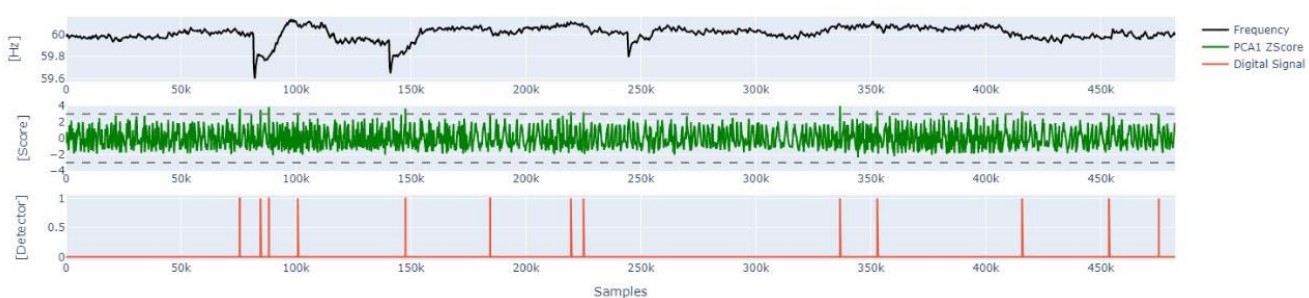

**Figure 20.** Event 2: detector event of PMU hydroelectric power station Ney Braga.

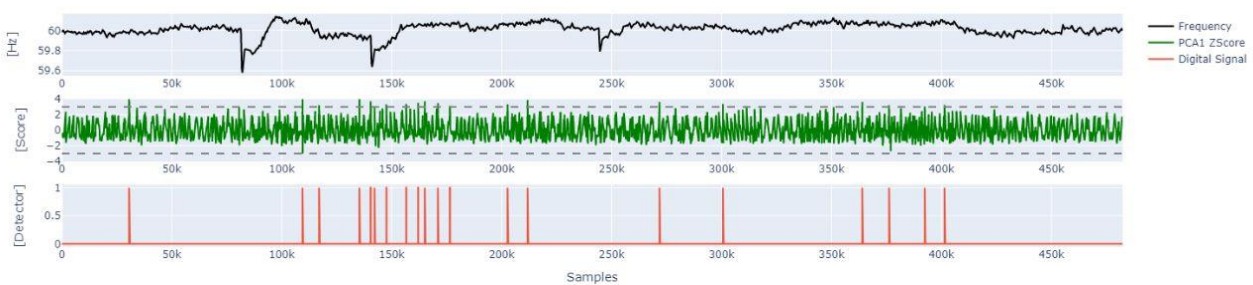

**Figure 21.** Event 2: detector event of PMU hydroelectric power station Salto Santiago.

The PCAs standardized are calculated including measures of frequency, rate of change of frequency, magnitude and angle of voltage and current of each PMU.

After the application of Step 1 to 8 (as described for Event 1), Figure 22 shows the already-detected points of each PMU (Step 7) and the final points detection after the application of the Pearson correlation (Step 8).

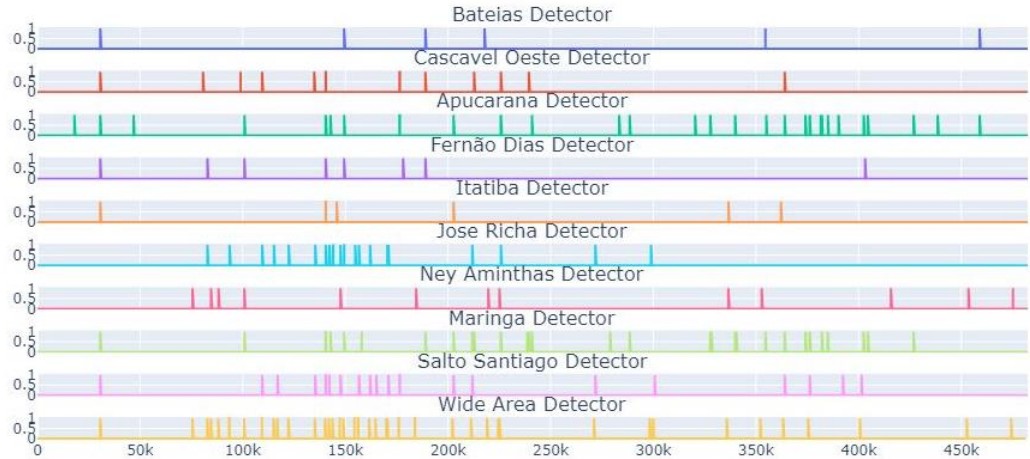

**Figure 22.** Event 2: PMUs detectors and final detector.

Figure 23 shows the final detection (Step 9) together with a signal of frequency to facilitate the comparations and analysis.

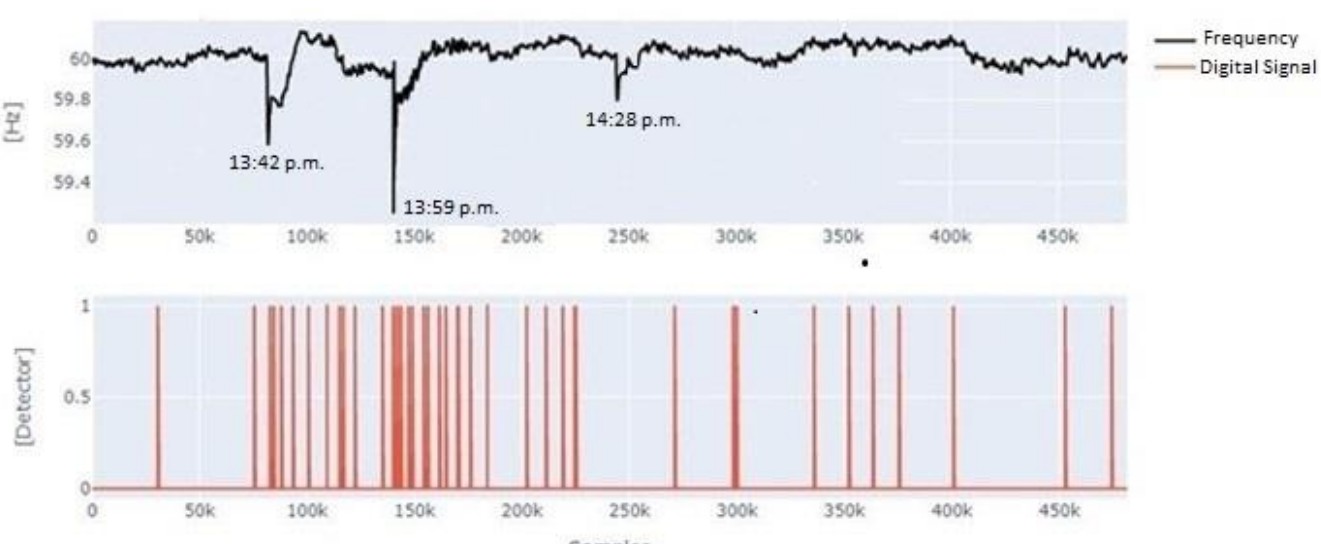

**Figure 23.** Event 2: final detector.

Figure 22 shows the detections realized by each of the nine PMUs of Apucarana, Cascavel, Bateias and Maringa, Itatiba, Araraquara, José Richa, Ney Braga and Salto Santiago captured during Event 2 (data from 1:20 p.m. until 3:34 p.m.).

According to Figure 22, the first anomaly, that occurred at 1:42 p.m., was detected by seven PMUs (Bateias, Cascavel, Apucarana, Fernão Dias, Itatiba, Maringá and Salto Santiago) that are installed inside the western region of Paraná, where Event 2 began. The PMU Apucarana was more sensible because it is in the western region of Paraná, where the anomaly began. In Figure 23, can be observed the final detection was found at 13:42 p.m.

The second anomaly occurred at 1:59 p.m. According to Figure 22, it was detected by seven PMUs (Cascavel, Apucarana, Fernão Dias, Itatiba, José Richa, Maringá and Salto Santiago). In Figure 23, it can be observed that the final detection was found at 13:59 p.m.

At 2:28 p.m., it was not detected anomalous at any of the PMUs neither by the final detection (Figure 23). After this time, ten false detections occurred.

If only Step 2 was implemented (identification of measurements outside the operational limits), none of the anomalies in the second event would have been detected and stored. The method proposed in [15], which uses the isolation forest, K-means and LoOP techniques, was not able to detect the event illustrated in this work.

The threshold settings of three deviations of Step 7 can be changed and optimized depending on the characteristics of the electrical power system. These were configured as the literature suggests, so the detection sensitivity can be increased as a function of fine tuning.

## 5. Conclusions

This work is result of a research and development project focused on the use of synchronized phasor measurement data.

The solution developed involves the combination of technologies and software engineering to handle large volumes of data. The platform used integrates several functionalities and supports the operation of the electrical network and the maintenance of the synchronized phasor measurement system.

To create a database with a history of occurrences of the national interconnected system of Brazil, and to keep records for long periods of time (years), a method was implemented to detect and store only the events. It uses principal component analysis and Pearson correlation.

The method implemented was able to detect and store important events that occurred during the operation of the national interconnected system of Brazil with good results and includes:

- status detection made possible by algorithm embedded at PMUs that generates information about the device's status, which may indicate data error, PMU error, modified data, loss of satellite communication, identify numerical problems such as absence of values (Not a Number—NaN), null values among others. The purpose of this functionality is to monitor the status of the devices, account for reported situations and generate a report indicating potential PMUs for maintenance
- detection of numerical non-conformities as values outside the range of operation
- systemic event detection: a set of techniques to monitor measurement values and detect violations of operational thresholds.

These set detections automatically trigger data storage, enabling the maintenance of a history of electrical network occurrences.

Observing the results of the two events, Event 1 (which was much more impactful than Event 2) was well detected both by the technique proposed in [13] and by the one described in this article. While for Event 2, only the PCA technique was able to detect the event, suggesting that PCA can be robust enough to detect electrodynamic events from different PMUs.

**Author Contributions:** Investigation, V.T.P., N.E.M.B., V.F.M.J. and A.K.d.S.; methodology, V.T.P., F.L.G., C.H.d.C., B.M.M., R.R.d.A., V.F.M.J., A.K.d.S. and V.A.d.S.; project administration, A.R.A., M.P.R. and R.R.d.A.; resources, R.R.d.A.; supervision, M.D.T., T.S.P.F., F.A.G., R.R. and A.R.A.; writing—original draft, F.L.G., M.D.T. and T.S.P.F.; writing—review and editing, T.S.P.F., M.D.T., G.L.-T. and A.R.A. All authors have read and agreed to the published version of the manuscript.

**Funding:** This research was founded by Companhia Paranaense de Energia—COPEL Generation and Transmission S.A. research and technological development (RTD) program, through the PD-06491-0531/2019 project, regulated by ANEEL.

**Data Availability Statement:** Not applicable.

**Acknowledgments:** The authors thank the grant for Technology Development of CNPQ—National Council for Scientific and Technological Development within the Ministry of Science, Technology, Innovations, and Communications. The authors thank the support of CAPES—Brazilian Federal agency for Support and Evaluation of Graduate Education with the Ministry of Education of Brazil. The authors also thank COPEL Generation and Transmission S.A for the funding from the research and technological development (RTD) program, through the PD-06491-0531/2019 project.

**Conflicts of Interest:** The authors declare no conflict of interest.

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
