# Peer review of "Electrical Event Detection and Monitoring Data Storage from Wide Area Measurement System"

_energies, doi:10.3390/en16041713_

Round 1

Reviewer 1 Report

The article presents a Data Management System Architecture applied to a Phasor Measurement Unit  installed in the state of Paraná, Brazil.

The investigated DMS that detect and storage events using Principal Component Analysis and Pearson Correlation.

I have few comments and concerns that would be taken into consideration to enhance the quality of the article. The comments are summarized as follows:

1- The number of authors (17) looks un practical. If some people have participated in some auxiliary tasks  they can be acknowledged at the end of the article.

2- The major contribution of the article is not clear enough. Thus, authors should demonstrate the main contribution compared with existing methodologies and analytics techniques.

3- All detected events are assigned to the change in the frequency. However, other parameters should be monitored as well.

3- Sensitivity of the applied algorithms should be addressed. Rate or percentage of False detection should be analyzed as well.

Reviewer 2 Report

Dear Authors, thank you for sending your article, but there are some parts to correct and integrate.

1) check the layout of lines 44-45.

2) Standardize the paper with the same font and size. The references (14-15,16) have characters with different sizes. Check the paper and correct it.

3) the References are few and some very old. Check and add more recent references.

4) In the paper it is not clear how the "ERRORS IN PHASOR ESTIMATION" affects. Have you analyzed this aspect? How does it affect? What's the mistake?

5) In the year 2022, the technique "use of synchronized phasor measurement data using Pearson Correlation", is producing a large number of journal articles.

I don't see references to articles from the year 2022 and they haven't been specified well what's new? Have you improved anything? It is not clear in the paper.

6) A schematization of the electrical network and mathematical formulas are missing. There are only acquisitions and it is not clear where your algorithm was implemented or simulated.

Best Regards

Round 2

Reviewer 2 Report

Dear Authors,

thank you for the changes made to the paper, now it's fine.

The style of the references should be improved, as explained in the MDPI guides.

The paper is accepted without further modifications.

Best Regards